# Current Developments in Cellular Therapy for Castration Resistant Prostate Cancer: A Systematic Review of Clinical Studies

**DOI:** 10.3390/cancers14225719

**Published:** 2022-11-21

**Authors:** Christina Steinbach, Almas Merchant, Alexandru-Teodor Zaharie, Peter Horak, Maximilian Marhold, Michael Krainer

**Affiliations:** 1Internal Medicine I, Department of Oncology, Medical University of Vienna, 1090 Vienna, Austria; 2Department of Radiation Oncology, Medical University of Vienna, 1090 Vienna, Austria; 3Nationales Centrum für Tumorerkrankungen (NCT), 69120 Heidelberg, Germany

**Keywords:** mCRPC, immunotherapy, cellular therapy, dendritic cells, CAR-T, vaccines

## Abstract

**Simple Summary:**

Prostate cancer is one of the leading causes of cancer-related deaths among men in the United States and Europe. While conventional treatment options either have high toxicity profiles or are limited by the tumor’s ability to develop resistance, immunotherapy is specific in targeting malignant cells. It can also enable the immune system to adapt dynamically while the tumor evades the destruction process. Cellular Immunotherapy in particular is a promising approach; however, there are gaps in knowledge in the current literature on its application. To contribute to a wider understanding of cellular immunotherapy, specifically focusing on the chimeric antigen receptors (CAR) T-cells, their benefits and application in clinical settings, we conducted a comprehensive systematic review.

**Abstract:**

Recently, the development of immunotherapies such as cellular therapy, monoclonal antibodies, vaccines and immunomodulators has revolutionized the treatment of various cancer entities. In order to close the existing gaps in knowledge about cellular immunotherapy, specifically focusing on the chimeric antigen receptors (CAR) T-cells, their benefits and application in clinical settings, we conducted a comprehensive systematic review. Two co-authors independently searched the literature and characterized the results. Out of 183 records, 26 were considered eligible. This review provides an overview of the cellular immunotherapy landscape in treating prostate cancer, honing in on the challenges of employing CAR T-cell therapy. CAR T-cell therapy is a promising avenue for research due to the presence of an array of different tumor specific antigens. In prostate cancer, the complex microenvironment of the tumor vastly contributes to the success or failure of immunotherapies.

## 1. Introduction

Prostate cancer (PCa) is one of the leading causes of cancer-related deaths among men in the United States and Europe, as well as the leader of estimated new cases of non-cutaneous cancer in the United States [1].

Androgen deprivation therapy (ADT), with or without chemotherapy, is the standard of care in metastatic prostate cancer (mCRPC). ADT can either be achieved surgically by bilateral orchiectomy or by suppressing testosterone levels chemically with Gonadotropin-releasing hormone (GnRH) analogues or antagonists [2]. Despite a good initial response to ADT, a significant fraction of patients relapse [3]. Once the tumor continues to progress despite castration levels of testosterone, known as castration resistant prostate cancer (CRPC), the disease becomes incurable and is associated with poor prognosis, as well as bone, lymph node, liver and brain metastases (mCRPC) [4,5]. Four different classes of medical treatment options improve survival among patients with mCRPC. These include taxane-based palliative chemotherapy, androgen-signaling targeting inhibitors, Radium-223 dichloride, Lutetium-177-PSMA-617 and immunotherapy [6,7,8,9,10,11].

While conventional treatment options either have high toxicity profiles or are limited by the tumor’s ability to develop resistance, immunotherapy is specific in targeting malignant cells. It can also enable the immune system to adapt dynamically while the tumor evades the destruction process [12].

Immunotherapy includes the use of but is not limited to cellular therapy (also known as as adoptive cell therapy), monoclonal antibodies, vaccines and immunomodulators to target and eliminate malignant cells [13,14] The goal set by most cancer immunotherapeutics is to activate a population of effector T cells that can infiltrate tumors and cause specific lysis of cancer cells [15].

Cellular Immunotherapy in particular is a promising approach; however, there are gaps in knowledge in the current literature on its application. To contribute to a wider understanding of cellular immunotherapy, specifically focusing on the CAR T-cells, their benefits and application in clinical settings, we conducted a comprehensive systematic review.

### 1.1. Mechanics of Cellular Immunotherapy

Cellular therapy refers to administering living cells to the patient, either actively or passively. The dendritic cell vaccine is an example of active cellular immunotherapy. However, the deployment of T-cells such as tumor-infiltrating lymphocytes (TILs), engineered T-Cell receptors (TCR), natural killer cells (NKs) and modified T-cells with chimeric antigen receptors (CAR) is a mode of passive cellular immunotherapy [16,17].

#### 1.1.1. Dendritic Cell Vaccine

Currently, the Sipuleucel-T dentridic cell vaccine is the only cellular immune therapy approved by the U.S. Food and Drug Administration (FDA) for asymptomatic or minimal symptomatic metastatic CRPC. Sipuleucel-T is a therapeutic vaccine that activates the host immune system by using autologous peripheral-blood mononuclear cells (PBMCs), including antigen presenting cells (dendritic cells) for regulating tumor control. The cells are activated ex-vivo with PA2024, a recombinant fusion protein, consisting of prostatic acid phosphatase (PAP), a prostate antigen combined with the granulocyte-macrophage colony stimulating factor (GM-CSF), used as an immune-cell activator [9,18]. Its approval was based on the efficacy demonstrated by results from the large phase III, double-blind placebo-controlled IMPACT trial, which showed prolonged overall survival (OS) by 4.1 months in the treatment group receiving Sipuleucel-T when compared to the placebo group (*p* = 0.03). Interestingly, there was no significant difference in progression-free survival (PFS) or time to clinical progression, as a former smaller phase III trial had already suggested [9,19].

#### 1.1.2. T-Cell Therapies

T-cells play an important role in the adaptive immune system and can therefore be used to enhance the patient’s immune response in different ways by using tumor-infiltrating lymphocytes (TILs-) in adoptive cell therapy or by using engineered T cell receptors (TCR) to boost the antitumor immune response or natural killer cells (NKs). TILs are collected from patient’s tumor, expanded ex-vivo in the presence of recombinant IL-2, to improve the anti-tumor cytotoxic function showing objective tumor shrinking in several types of cancer including metastatic melanoma [20]. Besides the aforementioned TILs-, modified T cell receptors are a novel class of molecules re-activating T-cells against cancer. By engineering T cells to express tumor antigen-specific receptors targeting intracellular peptides expressed on MHC-molecules, T-cells could increase antigen responsiveness with higher proliferation and cytokine production rates [21,22].

Natural killer cells (NK) play an essential role in tumor immunosurveillance. Preclinical as well as clinical studies have shown that diminished NK activity is associated with higher cancer frailty and metastatic burden. Adoptive NK cell transfer have been demonstrated anti-tumor activities in clinical use of hematologic cancers as well as in preclinical xenograft mouse models of solid tumors (glioblastoma, ovarian cancer, metastatic colorectal cancer) [23].

### 1.2. CAR T-Cell Therapy

In CAR T-cells, gene transfer technology has been used to manufacture T-cell receptors targeting specific tumor antigens to eliminate cancer cells [24]. CAR is a chimeric recombinant molecule consisting of an extracellular tumor antigen-binding domain typically represented by a single-chain fragment variable (scFv), a transmembrane (CD3, CD8, CD28 and FcεRI) and an intracytoplasmic region composed of CD8, CD28 or CD137 and CD3ζ as an intracellular signaling domain [25,26]. Depending on the presence of one or more costimulatory molecules, CAR T-cells can be classified into four generations. First-generation CAR T-cells permit T-cell activation but failed to enable persistence of activated T-lymphocytes. Through additional costimulatory domains (CD27, CD28, CD134, CDB7), second-generation CAR T-cells were able to persist in circulating blood. To further extend T-cell activation, a second costimulatory receptor was then added, resulting in third generation CAR T-cells (expressing CD28, 4-1BB). Whether the third generation of CAR T-cells is superior to the second generation is a matter of ongoing research.

The fourth generation of CAR T-cells, also referred to as “next-generation” or “armed”, increases the potency of anti-tumoral activity by the addition of proinflammatory cytokines (IL-12, IL-15, IL-18) or costimulatory elements such as knock-in (TRAC, CXCR4) or knock-out (PD-1, DGK) genes (Figure 1).

Patients peripheral blood mononuclear cells are obtained using leukopheresis, following T-cell isolation and activation via CD3/CD28. They are then genetically engineered to generate chimeric antigen receptors on their cell surface targeting cancer-specific antigens. Prior to CAR T-cell transfer, the patient undergoes lymphodepletion using fludarabine or cyclophosphamide, to further enhance immune responses [27,28,29].

### 1.3. Target Proteins

In an effort to identify potential immunotherapeutic targets for the treatment of PCa with CAR T-cells, three antigens, namely Prostate-specific membrane antigen (PSMA), Prostate stem cell antigen (PSCA) and Epithelial cell adhesion molecule (EpCam CD326), are currently of clinical interest and part of clinical studies.

#### 1.3.1. PSMA

PSMA is a transmembrane glycoprotein expressed in primary PCa [31], as well as lymph node and bone metastases [32]. PSMA expression is highly specific for PCa and correlates with high grading, castration-resistance and metastasis. PSMA has been investigated as a potential target for radioligand, bispecific T-cell engager (BiTEs) or CAR T-cell therapy [33,34]. Over the last two decades, PSMA has been investigated as a promising target, showing antitumoral cytotoxic T-lymphocyte response by HLA-A2 restricted PSMA-derived peptides [35,36,37,38], while other in-vitro and in xenograft models studied its immunotherapeutic potential as effective antibody targeting PSMA-expressing prostate tumor cells [39,40,41,42].

#### 1.3.2. PSCA

PSCA is a glycoprotein located on the cell surface of normal prostate tissue and in over 80% of PCa cells. Its expression correlates with high Gleason grading and the presence of metastatic disease [43]. PSCA has further been found to be expressed in other cancer types such as bladder-, pancreatic or gastric cancer [29,44]. PSCA-based vaccines have been shown to delay tumor growth and induce immune responses mediated by cytotoxic T lymphocyte activity, cytokine production and MHC (major histocompatibility complex) class I expression in the transgenic adenocarcinoma of the mouse prostate model (TRAMP) [45,46]. It is worth noting that Morgenroth et al. generated a PSCA-specific chimeric T-cell receptor causing cytotoxicity against PSCA-positive tumor cells in mice [47].

#### 1.3.3. EpCAM

EpCAM (CD326) is an epithelial cell adhesion molecule expressed by the surface of several solid tumors, including prostate-, colorectal- or lung cancer [48]. CD326 mediates specific intercellular cell-adhesion and is involved in cell signaling, migration, proliferation and differentiation and was shown to be highly expressed in rapidly proliferating tumors [49]. Deng et al. constructed an EpCAM-specific CAR and investigated its therapeutic potential in xenografts using the human PCa cell lines PC3 and PCM3. Interestingly, they found that treatment with this CAR resulted in tumor inhibition and prolonged survival in vivo [50].

## 2. Methods

We performed a systematic review in accordance with the PRISMA (Preferred Reporting Items for Systemic Reviews and Meta-analysis) guidelines (48) including studies published in the English language since January 2015. We searched the websites “ASCO Meeting Library”, “ClinicalTrials.gov” and “PubMed” for clinical trials testing autologous immunotherapeutics, including dendritic and CAR T-cell therapy in PCa. The following keywords and combinations were used according to Medical Subject Heading database (MeSH): “autolog” OR “autologeous” OR “autologic” OR “autological” OR “autologous” OR “autologously” AND “cell and tissue-based therapy” OR “cell” AND “tissue based” AND “therapy” OR “cell and tissue-based therapy” OR “cell” AND “therapy” OR “cell therapy” AND “prostatic neoplasms”. Furthermore, a filter naming clinical trials only was used. We then searched Pubmed by using the following MeSH terms concentrating on CAR T-cell therapy in PCa: “car-t” AND “cells” OR “cells” OR “cells” AND “prostatic neoplasms” OR “prostatic” AND “neoplasms” OR “prostatic neoplasms” OR “prostate” AND “cancer” OR “prostate cancer”. All references of the retrieved articles were checked to search for additional studies that we may have missed during the initial search. Reviews, preclinical trials and case reports were excluded. Only studies following Good Clinical Practice (GCP) Guidelines were eligible. Two co-authors (MM, BS) independently searched the literature and characterized the results. Figure 2 shows the process of collecting data.

## 3. Results

Our search yielded 185 results. After applying the exclusion criteria and excluding duplicates, 28 records were considered eligible. Sixteen out of 28 clinical trials studied dendritic cell therapeutics, and the remaining 10 studies examined CAR T-cell therapy.

The 13 clinical trials with clinical data available studied six immune cell-based therapeutics (total of 8 trials), namely Sipuleucel-T, BPX101, DCVAC/PCa, Tn-MUC1-dendritic cell vaccine as well as PSMA and PSCA-targeting CAR T-cells. The most common endpoints used in trials included in our study were safety endpoints, antitumoral activity, immune response, overall survival (OS), progression-free survival (PFS), overall response rate (ORR) and prostate specific antigen (PSA) response. The Results section has been categorized into two parts: (1) Dendritic Cell Therapy, describing trials involving Sipuleucel-T, BPX101, DCVAC/PCa, Tn-MUC1 (Table 1); and (2) CAR T-Cell Therapy, mapping trials focusing on PSMA, PSCA and EpCam CD326 (Table 2).

### 3.1. Dendritic Cell Therapy

#### 3.1.1. Sipuleucel-T

Due to signals of improved survival and low toxicity reported for Sipuleucel-T, the development of immunotherapies for PCa using autologous antigen presenting cells has progressed substantially in recent years [9].

PROCEED (NCT01306890) is a multicenter, open-label observational registry study including 1976 patients with asymptomatic or minimal symptomatic mCRPC who had received Sipuleucel-T. The study was conducted at urology and oncology centers as well as in academic sites and private practices between 2011 to 2017. Its endpoints consisted of OS, serious adverse events (SAEs), cerebrovascular events (CVEs) and anticancer interventions (ACIs). Median OS was 30.7 months with a median follow up of 46.6 months. SAEs of 3.9% and CVEs of 2.8% were reported. It is worth noting that 77.1% of patients received one or more anticancer interventions (e.g., chemotherapy, abiraterone/enzalutamide, radium-223) [51].

Results from two smaller, randomized phase II trials (NCT01807065, NCT01431391) investigating the use of Sipuleucel-T after sensitizing radiation therapy and combinational therapy of Sipuleucel-T and ADT have been published since 2015. In the first trial mentioned, the assumption that radiotherapy might act synergistically in mCRPC patients treated with Sipuleucel-T was not confirmed [52]. The second study examined whether the sequence of administration, i.e., Sipuleucel-T first and then ADT or vice versa, had an effect on PA2024-specific T-cell response in men with hormone-sensitive, non-metastatic, biochemically recurrent PCa. Sixty-eight patients were randomized in a 1:1 ratio to receive either Sipuleucel-T following ADT or ADT following Sipuleucel-T. ADT was given for one year. Both groups showed no difference in the time to PSA recurrence; however, PA2024-specific humoral response correlated with a longer time to PSA progression. An increased antitumoral immune response was observed for patients receiving Spiuleucel-T first. There is a need to further investigate whether this observation has clinical impact on patient prognosis [53].

NCT01981122 evaluated if PA2024-specific T-cell response differs in patients receiving Sipuleucel-T given concurrently with Enzalutamide or administered sequentially. The concurrent arm showed better T-cell response, although mortality did not differ in both arms [54]. Lastly, based on the hypothesis that improved survival was observed in patients with increased cancer specific immunoglobulin titers, a small phase I trial (NCT01832870) investigated the combination of Sipuleucel and Ipilimumab in patients with CRPC. The patients did show an increased immunoglobulin specific for PA2024 protein and PAP without increased adverse events occurring [55]. Aside from the aforementioned trials who had reported data, Table 1 summarizes ongoing trials investigating the safety and efficacy of dendritic cell therapy in patients with PCa.

#### 3.1.2. BPX101

BPX101 is a second generation, PSMA- targeting autologous dendritic cell (DC) based vaccine using inducible human CD40 as a co-stimulatory molecule to permit controlled DC activation. In a phase I (NCT00868595) trial published in 2017, immune upregulation and anti-tumoral activity in mCRPC patients treated with BPX101 were observed. Secondary endpoints defined as tumor tissue infiltration with CD4+, CD8+ T-cells and CD20+ B-cells, lymphocytic response measured through cytokine concentrations (e.g., NFα, IFN-γ, RANTES) and antibody response (e.g., IL-10, IL-6) in patients’ blood, prostate and skin biopsies were positive. In the study, 18 patients with a maximum of one prior chemotherapy received different doses of BPX101 followed by rimiducid, a membrane-permeable activating dimerizer drug for specific dendritic cell activation. Interestingly, no dose-limiting toxicities were observed. Antitumoral activity was evaluated by PSA decline, objective tumor regression and robust efficacy of post-trial therapy. A third of the patients progressed or died during follow-up. Longer median OS was observed in patients who had not received prior chemotherapy (530 days vs. 304.9 days in the post-docetaxel group) [56,57].

#### 3.1.3. DCVAC/PCa

A single-arm phase I/II trial (EudraCT 2009-017259-91) involving 27 patients with rising PSA levels after primary prostatectomy or salvage radiotherapy examined the use of autologous dendritic cells pulsed with LNCaP cells (DCVAC/PCa) following Imiquimod to support DC accumulation. The use of DCVAC/PCa significantly prolonged PSA doubling time (PSADT) from 5.7 months prior to immunotherapy to 18.9 months after 12 cycles (*p* < 0.0018), with no significant side effects recorded. Long-term follow-up will reveal if the observed prolongation of PSADT will translate into favorable patient outcomes [58]. In a different setting (EudraCT 2009-01295-24), combination chemoimmunotherapy with dendritic cells was given to mCRPC patients. The underlying rationale was the hypothesis that chemotherapy may neutralize tumor-triggered immunosuppression and thus enhance the effect of concurrent immunotherapy. Therefore, docetaxel was added to DCVAC/PCa application (on day 0, 14, 28, every 6 weeks). Results showed an increase in OS through the addition of DCs, with no serious DVAC/PCa-related adverse events observed. Although significant decreases of regulatory T-cells (TREGs) in peripheral blood were described, no immunological parameter significantly correlated with the survival benefit observed [59]. It is worth noting that Kongsted et al. investigated whether the additional use of autologous dendritic cell-based vaccines would induce the immune response in mCRPC patients treated with docetaxel in a randomized phase II study (NCT01446731). In this trial and in contrast to the aforementioned results, PSA response was reported to be similar in both groups (docetaxel vs. combinational therapy, 58% vs. 38%, *p* =0,21). In addition, no improvement in PFS (progression free survival) or DSS (disease specific survival) was described [60].

#### 3.1.4. Tn-MUC1

MUC1 is an epithelial glycoprotein, which was shown to be hypoglycosylated and overexpressed in a variety of solid tumors, including PCa. Scheidl et al. (NCT00852007) evaluated the safety of the Tn-MUC1 dendritic cell vaccine in a non-metastatic CRPC (nmCRPC) patient cohort. In this trial, Tn-MUC1-loaded DCs appeared to be safe and showed biological activity as well as T-cell response in nmCRPC patients [61].

### 3.2. CAR T-Cell Therapy

Despite highly promising preclinical data, CAR T-cell therapy has shown limited efficacy in PCa patients. Recently, three surface epitopes of interest, namely PSMA (prostate-specific membrane antigen), PSCA (prostate stem cell antigen) and EpCAM (epithelial cell adhesion molecules), were used as targets in clinical trials of CAR T-cell-based therapy in PCa. Seven of ten clinical trials reviewed in this article targeted PSMA, whereas two used PSCA as their surface epitope of interest. One trial targeted EpCAM. Among the ten phase I/II trials mentioned above, three had reported early results before completion of the study. The most common endpoints were safety, dose finding and antitumoral efficacy.

#### 3.2.1. PSMA

The first trial (NCT00664196) using first-generation anti-PSMA CARS showed negative results with poor CAR T-cell persistence and inferior efficacy unlike upcoming next-generation therapeutics. Two out of five patients treated showed partial responses as measured by reduction of PSA levels in peripheral blood [70,71]. In a second phase I dose-escalating study (NCT01140373), a second-generation CD28-based, anti-PSMA CAR was used, demonstrating efficacy and stabilization of disease in two out of four patients in cohort one. In cohort two, a higher number (dose) of CAR T-cells was administered. Observations made included increased interleukin levels (Il-4, Il-8, IP-10, sIL-2ra, IL-6), intermitted fever spikes and CAR T-cell persistence for up to 2 weeks [72].

One particular promising approach is the use of PSMA-specific TGFß-receptor dominant-negative autologous CAR T-cells to help facilitate continuous proliferation of CAR T-cells and enhance their antitumoral potential by suppressing the tumor microenvironment. TGF-ß (transforming growth factor beta) acts as an important promotor of tissue growth with proangiogenic and immunosuppressive effects on the tumor microenvironment and functions as a mediator of metastasis by increasing the expression of tissue-specific metastasis genes, all of which promote cancer progression [73,74]. Previously used mouse models demonstrated enhanced potency of CAR T-cells by the co-expression of dominant-negative TGFß in PSMA-directed CAR T-cells compared to unmodified CAR T-cells with increased immune responses, long-term in-vivo persistence and tumor eradication [75]. Another preclinical trial, using PSMA-specific TGFRß-receptor dominant-negative autologous CAR T-cells, improved the antitumoral ability of CAR T-cells and intensified the immune suppressive response to TGFß [76].

Based on these preclinical observations, a single center, single arm phase I clinical trial (NCT03089203) was initiated, assessing feasibility, tolerability and efficacy of PSMA-redirected/TGFß-insensitive CAR T-cells in metastatic CRPC treatment. Cohort 1 (single dose of 1–3 × 10^7^/m^2^) and cohort 2 (1–3 × 10^8^/m^2^) were completed without observing dose limiting toxicity; however, a reversible cytokine release syndrome responsive to tocilizumab was described. Of note, in cohort 3 of the trial, CAR T PSMA-TGFßRDn cells were administered following lymphodepleting chemotherapy (cyclophosphamide/fludarabine). Results of this cohort, however, have not been published to date [77]. Furthermore, an ongoing multi-center study, NCT04227275, aims to expand clinical knowledge of CAR T-cells, including dose escalation, safety, preliminary efficacy and feasibility [78]. Table 2 shows ongoing studies investigating the safety and efficacy of anti-PSMA CAR T-cells in patients with PCa.

#### 3.2.2. PSCA

There are currently two trials targeting PSCA as tumor antigen in solid tumors. To evaluate the feasibility, safety and clinical activity of PSCA-targeting CAR T-cells (BPX-601), a phase I/II open-label trial (NCT02744287) with 151 patients with previously treated, advanced PSCA-expressing solid tumors (pancreatic and prostate) was initiated [82]. Participants received BPX-601 followed by single or multiple infusions of rimidicid until a recommended cell dose was reached. When initiated, phase II will assess safety, pharmacodynamics and activity of BPX-601. The data of nine patients with pancreatic cancer who have completed phase I showed promising results with enhanced T-cell expansion and prolonged persistence of BPX-601. Four out of nine patients reached stable disease ≥8 weeks. Clinical data on PCA patient treatment were yet to be reported [83].

Lastly, a different phase I, open-label, study (NCT03873805) identified through our search, which also uses second generation PSCA CAR T-cells containing an intracellular 4-1BB co-stimulatory domain, lists study side effects and describes best dose of second-generation PSCA-CAR-4-1BB T cells in four cohorts (1,1b,2,3), each receiving different doses of anti-PSCA CAR T-cell therapy in patients with mCRPC (*n* = 33) from August 2019 till December 2023. Early results showed that five participants successfully manufactured PSCA-BB ζ cells, and that three patients completed cohort 1–100 million (M) CAR T × 1 alone (without lymphodepletion) [84,85].

#### 3.2.3. EpCAM

Since Deng et al. developed EpCAM-specific CARs showing significant tumor growth inhibition and prolonged survival in mouse xenograft models using the PC3 human prostate cell line, the antigen became a new target of interest in cellular therapy [50]. Currently, one trial (NCT03013712) being conducted to evaluate safety and efficacy of EpCAM targeting CAR T-cells in the treatment of patients with EpCAM positive cancer, including prostate, plans to enclose 60 participants [86].

## 4. Discussion

In 2021, five anti-CD19 CAR-T-cell therapies received FDA approval for treatment of hematological malignancies. In particular, patients with acute lymphoblastic leukemia, where CD19 is known as a suitable marker, show high response rates, leading to their following FDA approval. The first multicenter, open-label, phase III trial comparing the safety and efficacy of bb2121 vs. standard treatment in refractory or relapsed myeloma with an estimated enrollment of 381 participants complete recruiting May 2022 [87,88]. Although CAR T-cell therapy has celebrated success with numerous of hematological tumor identities, it still faces numerous barriers in solid tumors, in particular in the treatment of prostate tumors.

According to a recent review, the epi- and intratumoral genetic heterogeneity in PCa is a major contributor to the failures faced by researchers and clinicians [29]. Till now, the current state of knowledge and the approval of the aforementioned CAR T-cell agents is based on selected phase I/II trials. Data from large, multicenter phase II/III studies are still pending and very much needed to further develop their impact in clinical practice.

High-grade PCa is characterized by low levels of tumor infiltrating lymphocytes and a poorly understood interaction between adaptive and innate immune response [89,90,91]. Further several targets are available, but with no comparable specificity in antigen expression for PCa.

In addition, the most prevalent metastatic site in PCa is the bone with a majority of patients, over 90% in some studies suffering from bone metastasis only [88].

To achieve durable remissions in CRPC, CAR T-cells must effectively migrate to the bone lesions, and attack in and survive in a mostly hostile and immunologically little understood microenvironment [92]. Improving tumor trafficking could be achieved in two main ways: (1) CAR T-cells with chemokine receptors secreted by PCa such as CCL2, CCL21, (2) by converting the “cold” anti-immunogenic PCa into “hot” immunogenic tumor cells, thereby engaging the intrinsic ability of PCa by employing chemotherapy or local radiotherapy alongside with CAR T-cell therapy [93,94,95,96].

Once a CAR T-cell reaches its intended target, it has to face the inhibitory tumor microenvironment with lack of nutrients, low pH levels, hypoxia and a high amount of immunosuppressive cells containing fibroblasts, Tregs, tumor-associated macrophages and myeloid-derived suppressive cells known to weaken the immune response at multiple levels and various antigen escape mechanisms resulting in tumor resistance [97].

In a preclinical mouse model of PCa, PSMA-CAR T-cells combined with a dominant-negative TGFß typ II receptor binding domain overcame the aforementioned problem and led to increased proliferation rates as well as long-term persistence in vivo [75]. Previous mouse models have shown that, by targeting Interleukin-4, an inhibitory cytokine expressed by the TME in pancreatic cancer, next generation anti-PSCA CAR T-cells were able to eliminate tumor cells [98].

By developing NK-92 cell lines with PSMA-recognizing CARs as a novel approach, a potent cytotoxic response was achieved through acquired lytic activity against PSMA-overexpressing PCa cells. In addition, tumor growth was slowed and an increased production of IFN-y levels was described, improving survival in vitro and vivo mice models [99].

To increase efficiency, additional therapeutic approaches could be explored in form of multimodal combination therapies of CAR T-cells with ADT, radiotherapy, chemotherapy or other immunologic targets [100]. By triggering apoptosis and hence the formation of apoptotic bodies that can be used by antigen-presenting cells to activate T cells, ADT could be used to sensitize the immunosuppressive microenvironment of PCa for CAR-Ts. In addition, increased CD4+ and CD8+ T-cell infiltration after androgen deprivation has been described in the literature [101,102,103]. Chemotherapeutics such as cyclophosphamide or fludarabine have been used for depletion of endogenous lymphocytes before administration of CAR T-cell therapy, with the result of a reduced number of Tregs, anti-CD19 CAR T-cell expansion as well as longer persistence resulting in an increased anti-tumor immune response [104,105]. Furthermore, chemotherapy could enhance CAR T-cell penetration through pre damaged tumor cells and lead to enhanced cytokine release in TME, resulting in increased CAR T-cell activation [100]. To our knowledge, one open-label, multicenter phase 1b/2 study (NTC03910660) currently investigates the safety, tolerability and efficacy of a combined therapeutical approach with BXCL701 and the immune checkpoint Pembrolizumab in 40 mCRPC patients [106]. Future studies are required to determine whether the combination of CAR T-cells with chemotherapy, ADT, radiotherapy or other forms of immunotherapy could have clinical utility in patients with PCa.

## 5. Conclusions

While CRPC was the first solid tumor where a dendritic cell therapy could show an improvement in survival, clinical research with CAR T-cell therapy in this patient group is in its infancy compared to lymphoma and even other solid tumors. The lack of specific antigens and the complex immunologic tumor environment in bone metastasis proves to be a continuing challenge. Ongoing clinical research is therefore warranted to lead to improving patient outcomes.

## Figures and Tables

**Figure 1 cancers-14-05719-f001:**
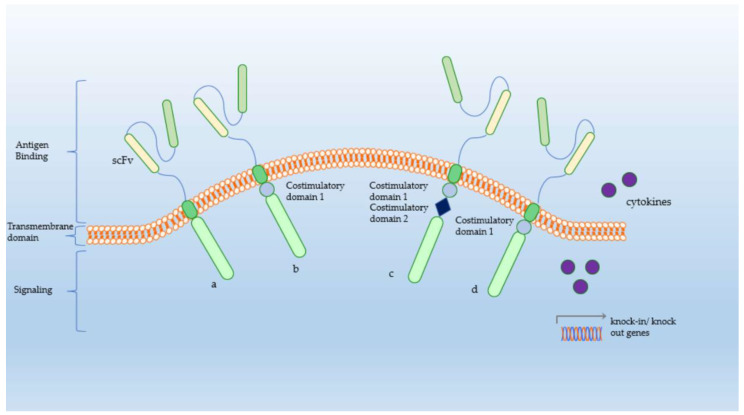
Generations of CAR-T Cells. (a) 1st Generation CAR-Ts; (b) 2nd Generation consisting of an extra costimulatory domain; (c) 3rd Generation CAR-Ts with a second costimulatory receptor; (d) 4th Generation CAR-Ts—“next generation” with an addition of proinflammatory cytokines and costimulatory elements (knock-in/knock out genes) [30]. Figures originally created by C.S.

**Figure 2 cancers-14-05719-f002:**
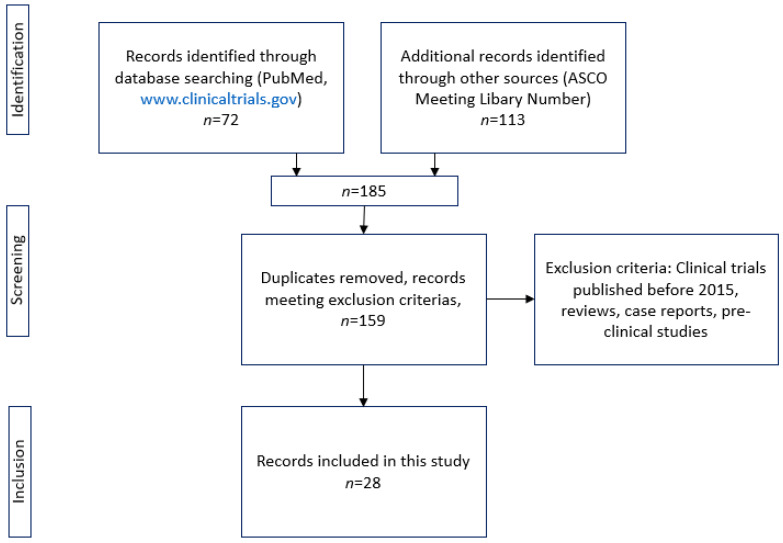
Methodology of literature search according to PRISMA guidelines. Figures originally created by C.S.

**Table 1 cancers-14-05719-t001:** Selected trials of emerging dendritic cell therapy in mCRPC. Tables originally created by C.S.

Identifier	Trial Name	Phase/Status	Endpoints
NCT04615845 [62]	Safety Evaluation of Autologous Dendritic Cell Anticancer Immune Cell Therapy (Cellgram-DC-PC)	phase I recruiting, *n* = 10 mCRPC	primary: measure CTCAE safety secondary: immune response, PSA effect
NCT02692976 [63]	Natural Dendritic Cells for Immunotherapy of Chemo-naive Metastatic Castration-resistant Prostate Cancer Patients	phase I *n* = 21 mCRPC	primary: immunogenicity of tumor-peptide loaded natural blood DCs Secondary: AE, PFS, QoL, PSA progression, OS, Time to opiate use, time to skeletal related events, ECOG, time to CHT, Radiographic PFS, feasibility
NCT02111577 [64]	Phase III Study of DCVAC Added to Standard Chemotherapy for Men with Metastatic Castration Resistant Prostate Cancer (VIABLE)	phase III completed, *n* = 1182 mCRPC	primary: OS secondary: PFS, PSA progression, duration of sceletal related events
NCT02105675 [65]	Phase II Study of DCVAC/PCa Added to Standard Chemotherapy for Men with Metastatic Castration Resistant Prostate Cancer	phase II completed, n = 60 mCRPC	primary: OS secondary: rPFS, duration of PSA progression, QoL, pain assessment
NCT01823978 [66]	Safety Study of BPX-201 Dendritic Cell Vaccine Plus AP1903 in Metastatic Castrate Vaccine Plus AP1903 in Metastatic Castrate	phase I completed, n = 19 mCRPC	primary: AE secondary: PSA, PFS, response to CHT after Immunotherapy, Reduction in circulating Tumor cells
NCT01487863 [67]	Concurrent vs. Sequential Sipuleucel-T & Abiraterone Treatment in Men with Metastatic Castrate Resistant Prostate Cancer	phase II completed, n = 69 mCRPC	primary: CD54 upregul. secondary: safety, immune response, Sipuleucel parameter
NCT02237170 [68]	Immune Monitoring on Sipuleucel-T (PROVENGE)	observational completed n = 36, PCa	primary: change in TREGsSecondary: changes in APC, cytokines, PSA- Specific immune resp. RNA transcript.-based sign.
NCT01706458 [69]	Provenge with or without pTVG-HP DNA Booster Vaccine in Prostate Cancer	phase II completed, *p* = 18, PCa	primary: OS secondary: number of circulating tumor cells, PAP-specific Antibody and T-cell Immune Responses

**Table 2 cancers-14-05719-t002:** Selected trials of emerging anti-PSMA-CAR T-cell therapy in CRPC. Tables originally created by C.S.

Identifier	Trial Name	Phase/Status	Endpoints
NCT04429451 [79]	Phase I/II Clinical Trial of 4SCAR-PSMA T Cell Therapy Targeting PSMA Positive Malignancies	Phase I/II enrolling, n = 100 PSMA positive tumors	primary: toxicity, adverse events, secondary: ORR, OS, expansion/persistence of 4SCAR-PSMA T cells
NCT04053062 [80]	A Phase I Study to Evaluate the Safety and Efficacy of PSMA-CART Co-expressing LIGHT in Treating Patients with Castrate-Resistant Prostate Cancer (CRPC)	Phase I recruiting, n = 12 CRPC	primary: toxicity, safety secondary: PSA radiographic response, duration time of CAR-T cells in-vivo
NCT04249947 [81]	A Phase 1 Dose Escalation and Expanded Cohort Study of P-PSMA-101 in Subjects with Metastatic Castration-Resistant Prostate Cancer (mCRPC)	Phase I recruiting, n = 40 mCRPC	primary: safety, dose finding, ORR

## Data Availability

The authors confirm that the data supporting the findings of this study are available within the article.

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
