# Peer review of "Current Developments in Cellular Therapy for Castration Resistant Prostate Cancer: A Systematic Review of Clinical Studies"

_cancers, 2022, doi:10.3390/cancers14225719_

Round 1

Reviewer 1 Report (Previous Reviewer 1)

All the recommended changes have been addressed.

This manuscript is a resubmission of an earlier submission. The following is a list of the peer review reports and author responses from that submission.

Round 1

Reviewer 1 Report

Dear Authors,

The review may be benefit form an additional section on NK CAR as well as a diagram representing the different generations of CAR T cells and checked for spelling and minor grammar. I have made some comments in the document itself for the authors to consider. The authors may wish to look at two clinical trials which I have found which do not appear to have been mentioned in this review. 

Author Response

Thank you for your valuable comments and suggestions. 

Spelling and grammar have been revised again. An additional section on NK CARs has been added to the Discussion. We added the two mentioned clinical trials in the review, thank you for the suggestion. A diagramm on the 4 generations of CAR Ts has also been added. We carefully rephrased the sections in the flow text according to the suggested recommendation. We deleted the section on CAR-T on lymphoma in joint agreement that this section is not required for our review.

Reviewer 2 Report

Christina Steinbach et al. reviewed the current developments in cellular therapy for castration-resistant prostate cancer.

Points to be considered:

1)the rationale of why the authors came up with this review.

2) what is the information that is not exactly available that motivated the authors to come up with this information. what are the current caveats and how do the authors highlight the current research in answering them? if not they need to address in future directions.

3) the authors need to highlight what new information the review is providing to enhance the research in progress.

4) in the discussion, this reviewer personally misses some insights regarding novel concepts in pc: as is now well known, tumors grow and evolve through constant crosstalk with the surrounding microenvironment, and emerging evidence indicates that angiogenesis and immunosuppression frequently occur simultaneously in response to this crosstalk. accordingly, strategies combining anti-angiogenic therapy and immunotherapy seem to have the potential to tip the balance of the tumor microenvironment and improve treatment response. prostate carcinoma does not make an exception (please refer to PMID: 32131507 and PMID: 35222436 and expand).

5) Indeed, angiogenesis in metastatic castration-resistant prostate cancer (mCRPC) has been extensively investigated as a promising druggable biological process. Nonetheless, targeting angiogenesis has failed to impact overall survival (OS) in patients with mCRPC despite promising preclinical and early clinical data. 

Author Response

Thank you for taking the time to review our paper and provide constructive feedback. 

(Answer to Items 1-3) The rationale of the submitted review was to summarize the current state of research in the field of cellular therapy in prostate cancer. In our opinion, there have been only a few articles in this area, especially when it comes to systematic reviews. We aim to give an expansive overview on completed studies with available data and studies that are still in progress.

(Aswer to Items 4-5) Crosstalk with the surrounding microenvironment in prostate cancer is an essential topic to discuss as well as angiogenesis in mCRPC. The studies you have mentioned have shown promising druggable biological process, but in our opinion, this topic goes beyond the range of this paper and should be addressed in further articles.